# Breeding Population Dynamics of Relict Gull (*Larus relictus*) in Hongjian Nur, Shaanxi, China

**DOI:** 10.3390/ani12081035

**Published:** 2022-04-15

**Authors:** Qingxiong Wang, Chao Yang, Defu Hu, Hong Xiao, Dong Zhang

**Affiliations:** 1College of Nature Conservation, Beijing Forestry University, Beijing 100083, China; wqx546@163.com (Q.W.); ernest8445@163.com (D.Z.); 2Shaanxi Institute of Zoology, Xi’an 710032, China; chaoy819@xab.ac.cn

**Keywords:** relict gull, breeding population, breeding islets, water level, Hongjian Nur

## Abstract

**Simple Summary:**

Long-term population monitoring is critical for informing the management and conservation of endangered species and for understanding population dynamics in response to changes in habitat suitability. We investigated breeding pairs and breeding islets of the Relict gull throughout Hongjian Nur from 2001 to 2020. We also analysed correlations between the breeding population and the breeding habitat, aiming to understand how environmental factors affect the breeding population. Our results suggest that the breeding population and breeding islets have been in an unstable status, and they show a positive correlation. Habitat restoration can be used as a compensatory method for breeding population restoration in Hongjian Nur in the future.

**Abstract:**

Hongjian Nur is an important breeding and stopover area for the globally endangered Relict gull (*Larus relictus*). This is where the species was first found in 2000. The breeding population of this species was monitored over the long term by directly counting nests from 2001 to 2020 in Hongjian Nur, Shaanxi, China. Our results suggest that breeding pairs increased rapidly, from 87 nests in 2001 to 7708 nests in 2010; at this point, the breeding population accounted for more than 85% of the global total, and was at the highest value during the past two decades. Subsequently, breeding pairs continued to decrease dramatically and reached a minimum number of 2054 nests in 2015, approximately 70% less than at their peak. In view of this situation, breeding islets were restored in 2014 and 2017, and the breeding population recovered slowly. Due to the changing ecological environment, breeding islets showed the same instability as the breeding population. Our conclusions support previous research, highlighting the importance of water level.

## 1. Introduction

The distribution and population variation in breeding seasons are important for understanding population dynamics and conserving species [1,2], and habitat and food abundance play a crucial role in the population size and life history of a species [3,4]. However, breeding habitat and food abundance are susceptible to the impacts of climate change and anthropogenic activities, which can cause considerable changes in the distribution of breeding colonies and populations [5,6]. Especially for colonial breeding waterbirds in semi-arid and arid regions, the instability of breeding sites can easily cause declines in reproductive success, survival and colonial nesting, thereby affecting the size of the breeding population [7,8,9]. Therefore, long-term monitoring of the breeding population and breeding habitat would contribute to understanding important factors of habitat change and to developing effective conservation plans [10,11].

The Relict gull (*Larus relictus*) is one of the most threatened gulls in the world [12], and the global population has been estimated at 10,000–20,000 individuals [13]. In China, it is listed under the First Class State Protected Wildlife in China, and it is recognized as a vulnerable species on the China Species Red List [14]. Internationally, it is listed as vulnerable in the Convention on International Trade in Endangered Species of Wild Fauna and Flora (CITES) and the Convention on the Conservation of Migratory Species of Wild Animals (CMS) Appendices I [13,15,16,17].

The Relict gull is distributed around the Mongolian Plateau during the breeding season and breeds only on the islets of saline lakes in semi-arid and arid areas on the Ordos Plateau, the Gobi regions in Eastern Kazakhstan, the Midwest of Mongolia and the Russian Far East, and the largest colonies occurring in the Ordos Plateau in China are known as the Ordos subpopulation [18]. In September 2000, breeding colonies of the Relict gull were first found in Hongjian Nur, Shaanxi Province [19]. The breeding population and habitat were surveyed from 2001 to 2020, and the breeding population accounted for more than 50% of the global total for many years. Therefore, Hongjian Nur is the largest and most important breeding and stopover area [20]. However, Hongjian Nur has experienced increasingly significant lake shrinkage resulting from drought, agricultural irrigation, coal mines and an increase in the exploitation of groundwater resources since 1999 [21]. As a result, these factors have led to breeding habitat deterioration, which means a large size of the breeding population cannot be supported. In this case, in 2013, some breeding individuals began to move successively to new breeding sites, such as Kangba Noel National wetland park in Kangbao County, Hebei Province [22]; Ao-tai Lake in Hohhot city, Inner Mongolia [23]; Gouchi Wetland in Dingbian County, Shaanxi Province [24]; Shuangmaotou Lake in Yanchi County, Ningxia Hui Autonomous Region [25]; and HaotongChahan Nur in Wushenzhao town, Inner Mongolia [26].

Thus, our approach aims to understand breeding population dynamics and breeding islets transition during the past two decades. It also provides useful references and predictions for understanding the population and the potential distribution of a globally vulnerable species.

## 2. Materials and Methods

### 2.1. Study Area

This study was conducted in the Hongjian Nur National Nature Reserve (38°13′–39°27′ N, 109°42′–110°54′ E), Shaanxi Province, China (Figure 1). Hongjian Nur is a closed inland lake in a semi-arid and arid area, at an elevation of 1200 m, with seven rivers flowing into the lake in history. However, only one river has water inflow;the other rivers are basically dried up at present. According to the soil matrix of the islets, there are two types of vegetation: one type is a sparse sandy herbaceous community, which is composed mainly of *Artemisia silversiana*, *Salsola ruthenica*, *Stipa bungeana*, *Argophyron cristatum*, *Phragmites communis*, *Heteropappus altaicus*, *Puccinellia tenuiflora*, etc. The other type is red sandstone and shale soil, which will mostly be bare and contain a few *Carex rigescens* and *Salsola ruthenica*. The average annual evaporation is more than 1700 mm, and the average annual precipitation is less than 500 mm, together with drought, upstream closures and overexploitation of groundwater. These situations have always existed and have led to a drop in the groundwater level and thus affected the lake area and the water level of Hongjian Nur. Therefore, Hongjian Nur is currently a saltwater lake, with a pH value of 9.0. Meanwhile, the lake area has gradually shrunk from 50 km^2^ in 1999 to 31 km^2^ in 2015 [27,28], and the water level is continually falling. Consequently, the lake naturally forms many isolated, different sizes of islands in Honjian Nur, and these are called islets. During the study period, a total of 23 islets were recorded, and the area ranged from 0.06 ha to 5.4 ha.

There are more than 170 species of birds in this area, of which almost 80 species are waterbirds, accounting for more than 45% of the birds. Almost 10 species of waterbirds breed on the islets. There are three species of Laridae as sympatric Relict gulls, namely, the Brown-headed gull, the Gull-billed tern and the Common tern. The Relict gull is the largest breeding population for these Laridae;it is a colonial breeding waterbird on the islet.Nest sites selected bare or spare sand, and the shortest distance was over 2 m to the water edge.

### 2.2. Data on Relict Gull Breeding Population

Field work was conducted from May to July each year, and nests were counted with the direct counting method, with people using four-person kayaks landed to visit the islets. Because the Relict gull is monogamous, a nest is equal to a breeding pair. To gain accurate data for the nests, two people went to the islets at a time; one person marked the nests by drawing a circle, and the other person counted the nests. To minimise interference with the reproductive period, the optimal time for collecting data on the nests, clutch sizes and breeding sites was 07:00–09:00 from 22 May to 26 May every year. The temperature in a nest is maintained at 20 °C to 26 °C during incubation, but the temperature here varies greatly between the morning and afternoon in the semi-arid and arid area. Before 07:00, the temperature was below 20 °C; after 09:00, the temperature was higher than 26 °C. Therefore, unprotected hatching eggs were exposed for a long time, which would affect the hatching success at a low or high temperature. In addition, clutch laying in some nests was still incomplete, or replacement clutches were laid before 22 May, and some nestlings were hatched after 26 May, and then the nestlings of different nests clustered together. Therefore, this is a certain time period when the reproduction is most stable, and this is also the reason to minimize interference in the breeding population and to gain accurate counts of the nests and clutch sizes.

All breeding sites and islet locations were located on a map with a global positioning system (GPS, Unistrong G350), which facilitates measuring and understanding colonial distribution patterns with Unistrong GIS office software. Generally, the breeding habitat would be abandoned to breed for the Relict gull when some islets became connected to the lakeshore and formed peninsulas due to a continuous drop of water level. Therefore, in the winter of 2014, two available peninsulas were chosen and separated into two islets, and one was restored again in the winter of 2017. The breeding habitat was restored, and some breeding individuals reselected to breed on the restored islets.

### 2.3. Statistical Analysis

Nests, total eggs and clutch sizes were determined with the direct counting method, and all means are presented ± the standard error, with the range from minimum to maximum presented in brackets. We used a Kruskal–Wallis test to assess differences in clutch size, hatching success, fledging success and reproductive success between years. If a significant difference was found, then pairwise comparisons were made using a Mann–Whitney U test, and the correlation coefficient (r) was calculated with the size of colonial nests by Pearson’s test (*p* < 0.05) with SPSS software (Version 22.0, IBM 2013). All diagrams were drawn with SigmaPlot software (Version 13.0, SSI 2015). 

According to the calculation formulas, mean clutch size (MCZ) = number of eggs laid/total nests. Hatching success (HS) = (number of eggs hatched/number of eggs laid) ×100%. Fledging success (FS) = (number of chicks fledged/number of chicks hatched) ×100% [29].

## 3. Results

### 3.1. Breeding Population Dynamics

The annual mean breeding size was 3409 ± 447 pairs (range 87–7708 nests, *n* = 20 years), but we found that the breeding population exhibited unstable fluctuations during the past two decades (Table 1). Breeding pairs were first recorded in the field survey from May to June 2001, a total of 87 nests. In the first decadal census (2001–2010), the breeding population showed a rapidly rising trend, which reached the highest peak of 7708 breeding pairs in 2010, which was approximately 85% of the global total at the time. Although the breeding population appeared to slightly decline between 2008 and 2009, declining by approximately 769 and 1251 breeding pairs, respectively, this did not affect the overall rising trend.

In the second decadal census (2011–2020), the breeding population sharply declined, and the lowest breeding size was only 2054 pairs in 2015, which represented a decrease of approximately 73% in comparison with the breeding size in 2010. However, the breeding population slowly increased again through a restoration of the breeding habitat, to a breeding population size of 3581 pairs (Table 1).

### 3.2. Reproductive Success

The annual mean clutch size was 2.43 ± 0.04 eggs (range 2.09–2.83 eggs, *n* = 20 years) (Table 1), with no significant difference in the breeding population size (U = 6.000, *p* = 0.327 > 0.05). Clutches were typically 2–3 eggs, which accounted for 92.41% ± 4.03% (range 82.45–97.29%, *n* = 20 years) (Table 1) of the clutches. There were only a very small number of clutches with 1, 4 and 5 eggs present, at 7.59% ± 4.03% (range 2.71–17.55%, *n* = 20 years) of the clutches. Incubation by both adults begins with the first egg, and the average clutch required 25.36 ± 0.15 d (*n* = 63, range 24–28) of incubation.

The annual mean hatching success was 89.85%±0.04% (range 84.6–97.6%, *n* = 20 years) (Table 1), with no significant difference in the breeding population size (U = 5.500, *p* = 0.268 > 0.05). The fledging success of young birds was only 65.71% ± 0.03% (range 60.7–74.7%, *n* = 20 years) (Table 1), with no significant difference in the breeding population size (U = 7.000, *p* = 0.462 > 0.05). Our research found that the fledging nestling of clutch size was 1.43 ± 0.02 (range 1.26–1.80, *n* = 20 years), and mortality was high, accounting for about 40% of the total clutch size. 

### 3.3. Variation of Lake Area and Breeding Islets Transition

In general, the largest lake area of Hongjian Nur was about 60 km^2^ in 1969; before this, the area was generally increasing. From the 1970s to the 1990s, the fluctuation of the lake area was relatively stable. After 1999, the lake area showed a dramatic decline from 50.3 to 31.4 km^2^, dropping by 38% (Figure 2). Meanwhile, the water level has dropped from a depth of more than 10 m to only about 4 m. The correlation analysis showed that there was no significant correlation between the lake size and the breeding islets (r = −0.392, *p* > 0.05), but the lake area at 35–50 km^2^ was beneficial to the formation of breeding islets; any lake too large or too small would seriously affect breeding islets.

A total of 23 islets were investigated in Hongjian Nur during the past two decades (Figure 1), a place that has also fluctuated unstably in numbers and areas of islets. From 2001–2004, there was only one suitable islet in Hongjian Nur. After that, breeding islets showed a rapid increase from 2005 to 2010, during which time islets were at the highest peak of 9 in numbers in 2010. Accordingly, the breeding size also had the fastest growth rate, with an annual mean increasing more than 1000 breeding pairs. Then, breeding islets showed a rapid decline after 2011, and the lowest number of 3 suitable islets were recorded from 2015–2016; at this time, the breeding population was also at its lowest number (Figure 2). Although Hongjian Nur has increased from 5–8 breeding islets from 2017–2020, this is the result of the restoration of islets and a rising water level with river dredging and flooding in 2017. Without these factors, the breeding habitat could have deteriorated even further.

In general, the number of breeding islets showed a positive correlation with the breeding population size (r = 0.722, *p* < 0.01).

## 4. Discussion

The long-term monitoring of a bird population plays a crucial role in understanding population size and trend, and mastering important population parameters such as reproductive rate, mortality rate and life span, and it provides important data for further studies on the mechanism of environmental change in population dynamics [30,31]. Our research shows that the breeding population of Relict gull appeared in an unstable fluctuation in Hongjian Nur over the past two decades. Therefore, we speculated that the maximal size of the breeding population may be more than 15,000 breeding individuals in China from the perspective of the breeding population dynamics during the past two decades. However, there have been no reports on the status and distribution of the Relict gull from abroad since 2000; therefore, the breeding population of the globe is unclear.

The Relict gull showed an excellent population fertility in Hongjian Nur during the past two decades, in which the average of HS and FS were 89.8% and 65.7%, respectively. Therefore, the breeding pairs also increased with the increasing number of suitable breeding habitats, which is consistent with the first decadal census. Additionally, the breeding population of Hongjian Nur originally derived from Taolimiao-Alashan Nur, which was confirmed for the results of the Relict gull banded in Taolimiao-Alashan Nur and the recovery of banding information in Hongjian Nur [20].

The results showed that the fluctuation of the breeding population was correlated with the change of breeding islets. Due to a continuous decline of the water level, many suitable breeding islets are rapidly disappearing, leading to a severe degradation of the breeding habitat, which is unable to support a capacity of a vast breeding population. As a result, the breeding population has declined dramatically since 2011, to the lowest number of few more than 2000 breeding pairs in 2015. However, we believe that many breeding pairs failed to gain a breeding habitat, which can be unselected to disperse or migrate to breed in other locations. The case was confirmed after several new breeding sites were discovered since 2013 [22,23,24,25,26]. We believe that the changes in water level are a key factor affecting the breeding population, a statement that is consistent with previous studies in Taolimiao-Alashan Nur, Inner Mongolia [18].It has been proposed that avian species select breeding habitats based on factors related to successful nesting and the rearing of chicks [32,33], habitats that provide sufficient food and development and adequate cover for protection from exposure to severe weather and to hide from potential predators [34,35]. Islets were selected by this species for nesting, hatching and brood rearing in terms of safety and protection from predators and human disturbances. The Relict gull is a colonial breeding waterbird that expresses a strong defence system, which make it difficult for raptors to prey on them. In addition, the isolation of water makes it difficult for small mammals and human to land the islets, so it is safer to breed on the islets for this species. Similarly, the Relict gull will resolutely abandon a breeding site when the islet is connected to the shore of the lake and forms a peninsula. In view of this situation, when the peninsula was restored into an islet, this achieved a significant result, increasing the annual average by more than 1000 breeding pairs. Therefore, habitat restoration is an effective compensatory method to solve the problem of a vast breeding population and the increasing degradation of suitable breeding habitats.

In a word, in order to determine the global breeding population size of the Relict gull, it is necessary to establish a global monitoring network or a synchronous survey. In addition to the long-term monitoring of known important breeding distribution areas, such as Taolimiao-Alashan Nur, Hongjian Nur, Chahan Nur and Kangba Noel, potential or disappearing breeding sites should also be investigated.

## 5. Conclusions

In summary, our main findings were: (i) finding out the breeding population size of the Relict gull in Hongjian Nur over the past two decades; (ii) understanding how breeding islets were affected by a change in the water level, thus limiting the capacity of breeding pairs; and (iii) solving the shortage of breeding sites through a compensatory method of habitat restoration. Our conclusions support previous studies that suggest that environmental factors affect the breeding population dynamics of this species. Moreover, we suggest that it is necessary to examine the foraging habitat and food abundance for further study, in addition to the influence of the water level for breeding habitat.

## Figures and Tables

**Figure 1 animals-12-01035-f001:**
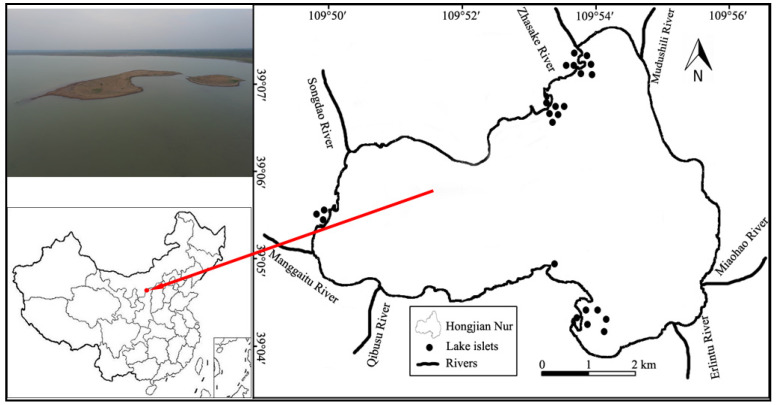
Map of the Hongjian Nur shows the breeding islets’ distribution, where relict gulls have bred at least once between 2001–2020. Upper left is aerial photo of islets;the dots show the location of the different islets within the breeding areas, and the red dot shows the location of Hongjian Nur, Shaanxi, China. (Schematic drawing based on GPS data of 2014.)

**Figure 2 animals-12-01035-f002:**
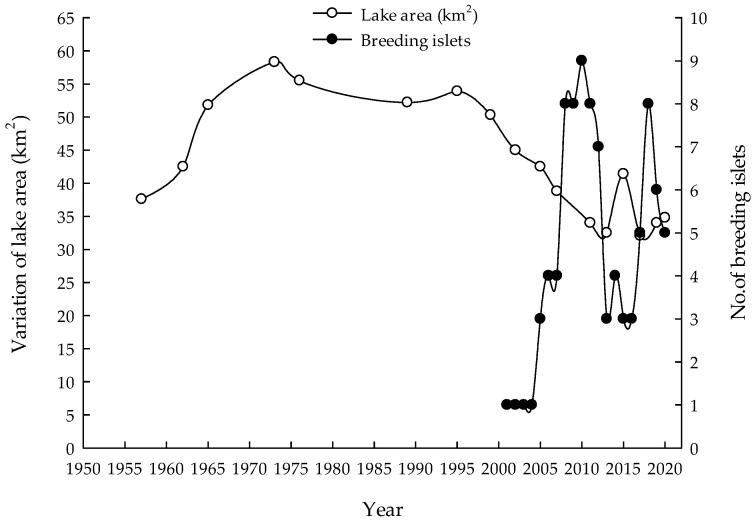
Relationship between lake area and breeding islets in Hongjian Nur over 60 years.

**Table 1 animals-12-01035-t001:** Numbers of breeding pairs, clutch size, reproductive success of Relict gull in Hongjian Nur, Shaanxi Province from 2001–2020.

Year	Clutch Size					Breeding Pairs (Nests)	MCZ	HS	FS
	1	2	3	4	5				
2001	5	64	15	3	0	87	2.18	85.1%	68.9%
2002	11	170	42	8	0	231	2.20	87.4%	66.3%
2003	22	792	858	24	0	1696	2.52	87.9%	61.0%
2004	61	1045	1267	36	0	2409	2.53	90.3%	62.4%
2005	72	962	1371	55	0	2460	2.57	91.2%	65.1%
2006	498	1761	700	26	0	2985	2.09	84.6%	65.9%
2007	779	2399	1816	42	0	5036	2.22	87.8%	67.8%
2008	334	1494	1896	61	0	3785	2.44	89.3%	60.7%
2009	447	1771	1958	88	3	4267	2.40	88.3%	63.3%
2010	581	3163	3920	43	1	7708	2.44	89.5%	62.4%
2011	627	2397	4515	61	4	7604	2.53	95.4%	69.6%
2012	559	1963	2562	52	4	5140	2.41	88.5%	63.2%
2013	310	1367	2820	47	1	4545	2.57	90.9%	69.2%
2014	464	1967	1919	23	0	4373	2.34	87.6%	64.2%
2015	228	708	1081	37	0	2054	2.45	89.9%	74.7%
2016	465	1102	1038	9	0	2614	2.23	87.4%	64.8%
2017	270	839	1606	36	3	2754	2.51	95.1%	65.9%
2018	112	286	2266	33	6	2703	2.83	97.6%	65.3%
2019	251	743	2507	68	12	3581	2.68	96.7%	67.3%
2020	195	728	1215	14	1	2153	2.49	86.4%	66.1%

## Data Availability

Not applicable.

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
