# Peer review of "Breeding Population Dynamics of Relict Gull (Larus relictus) in Hongjian Nur, Shaanxi, China"

_animals, 2022, doi:10.3390/ani12081035_

Round 1
Reviewer 1 Report
The authors describes an interesting long term study monitoring a population of nesting birds in a lake of China.
The authors discussed likely consequences of human disturbance, but did not describe the time and magnitude of those effects nor relates them in any way with their counting data.
Specific suggestions:
Please round up 10,000 to 20,000 in line43
Line 58, please provide a time reference since when coal mining, agriculture etc altered the study site.
Line 77 should be after the causes of water declines described in line 78 and 79.
Please, move upwards line 91 to 95 before line 77.
Explain why you can minimize interference with reproductive period selecting those dates and time of the day.
I cannot understand your explanation between line 104-108. Please rephrase it.
Line 110 and 112 Is this an explanation that relates to the justification of counting between 7-9 am?
Line 136 please add the time frame for this mean (20 years?)
Line 160. Is this a % of mortality?
Authors may explore the data of breeding pairs, clutch size and number of islet using a correlation analysis instead of a graphic expression in figure 2 and 3?
Author Response
Please consult “Reply to Review 1 Comments"

Reviewer 2 Report
This is an study focused on an interesting species due to its rareness and small distribution. The study relates the restorarion of breeding areas with breeding success.
The work is exhaustive and the number of year considered are enough to draw the authours' conclusions. The work demonstrates the breeding success that can be accomplished for this species when their breeding areas are restored.
I think the article would improve if authours did a description of the nest, and also the effects produced by the change in the water level in the nest need to be described more exactly. It is also needed a more detailed description of the intervention on the rivers to see how the water level is controlled.
It would also be interesting to see how the water level affects the arrival of predators to the eggs or chicks. The map is correct and sufficiently explanatory, but an aerial or satellite photo of the area and the islets would improve the understanding of the article, since for readers who don't know the area it is excessively abstract.
Author Response
Please consult "Reply to Reviewer 2 Comments"

Round 2
Reviewer 1 Report
Please consider to show in a graph a scatterplot and correlation between lake size and number of islets according to line 221.
Author Response
Please consult "Reply to Review 1 Comments (Round 2)"
